# MOTSC: Model-based Offline Traffic Signal Control

## Abstract

Urban areas are currently suffering from more and more severe traffic congestion. One of the most straightforward ways to relieve congestion is to optimize the control of traffic lights. Varieties of reinforcement learning (RL) methods are thus born and have shown good performance in traffic signal control. However, on one hand, the performance of the RL agents may be unstable due to limited interaction data in the early stages of training, leading to even more serious traffic congestion. On the other hand, most of the data generated by the interaction are discarded after training, leading to low data utilization. Hence, it is necessary to introduce offline reinforcement learning to traffic signal control, which trains RL policies without interaction between RL policies and the environment and fully utilizes the data collected in the past.

In this paper, we propose an offline traffic signal control method based on model-based offline reinforcement learning. We formulate offline policy optimization under traffic signal control and design the transition model. A theoretical proof has been given out that our method can estimate the state of out-of-distribution samples more accurately. We conduct extensive experiments to compare our method with methods of traffic signal control and offline reinforcement learning under offline traffic signal control, where our method achieves better performance on various metrics.

## 1 Introduction

Vehicle ownership per capita has increased with the development of modern urban traffic, leading to severe traffic congestion in urban area Mirchandani & Head (2001). Various studies are thus proposed to relieve traffic congestion. A significant and promising method among them is traffic signal control (TSC) Wei et al. (2019c), which improves transportation efficiency by optimizing the control plan of the traffic signal lights.

Many studies have been conducted on the topic of TSC. Researchers start using single-agent reinforcement learning (RL) to solve this problem Wei et al. (2018; 2019a); Zheng et al. (2019). Another group of RL-based studies models traffic signal lights as a graph in which agents cooperate with each other Wei et al. (2019b); Oroojlooy et al. (2020). These RL-based methods get distinctly better performance compared to transportation-based TSC.

While RL-based methods have enormous potential for TSC, there are some challenges to be aware of. First, the performance of the RL agents may be unstable due to limited interaction data in the early stages of training Ault & Sharon (2021). This undesired interaction between the environment and the RL traffic signal agents can lead to increased traffic congestion. Additionally, much of the interaction data generated during training is ultimately discarded, leading to low data utilization. Due to these problems, the deployment of RL-based TSC is sluggish.

In contrast to the scarcity of online transition data at the early training stage, there are plenty of offline transition data in the real world. These have prompted researchers to investigate policies that can be trained with RL in the offline dataset and then transferred to the real world.

Our idea to solve this problem is to make use of offline reinforcement learning (Offline RL) Lange et al. (2012); Levine et al. (2020). Offline RL does not require the interaction between RL policies and the environment and trains the policy only with offline data. This aligns well with our require-

ment to lower side effects in the real world and higher data utilization of offline data. Furthermore, the offline data are sampled from the environment by a trivial sampling policy. This idea fits the problem of TSC since real-world traffic takes trivial signal control policies (e.g. Fixed-time Miller (1963)) and produces enriched offline data.

However, it is challenging to effectively exploit offline RL in the problem of TSC. First, compared to other tasks in RL, TSC has more complex transition dynamics because each sample involves the dynamics of thousands of different vehicles, which is challenging to model Zheng et al. (2019); Oroojlooy et al. (2020); Wu et al. (2021a;b); Zhang et al. (2022). The complexity of the problem limits the performance of existing offline RL methods in TSC.

Another challenge is the tendency to overestimate the uncertainty. To keep the policy conservative, many offline RL methods Kumar et al. (2020); Yu et al. (2020; 2021) penalize the value function with an uncertainty term. However, this may grow uncontrollably in TSC due to the complexity of the traffic dynamics, leading to over-restriction on the exploration of the policy and finally the offline RL policy in TSC may act as the trivial sampling policy, bringing little improvement to traffic.

To tackle these two challenges, we proposed **M**odel-based **O**ffline **T**raffic **S**ignal **C**ontrol (MOTSC). We used a transition model as the fake environment to replace the real environment in the interaction with the RL policy. The transition model is specifically designed to adapt the traffic transition dynamics and separate one sample in offline data into 12 samples by the traffic movement, further improving the data utilization and greatly simplifying the transition model. This solves the first challenge of modeling complexity. Furthermore, the theoretical proof was given out that our transition model can reduce the overestimation of uncertainty, which makes progress in tackling the aforementioned second challenge. Extensive experiments were conducted to show that our method outperforms existing methods of both TSC and offline RL in the problem of offline TSC.

In summary, the main contributions of this paper include:

- In terms of the problem, we study and formulate the problem of TSC under the offline RL setting, which is essential for the application and deployment.
- In terms of the method, we propose MOTSC, a model-based offline RL algorithm for TSC to solve the problem of offline TSC.
- Extensive experiments are conducted and show superiority in training TSC policies under the offline setting.

## 2 PRELIMINARY

As the existing formulation of TSC Wei et al. (2019a); Zheng et al. (2019), we consider TSC as a Markov decision process (MDP) $M(\mathcal{S}, \mathcal{A}, \mathcal{T}, r, \gamma)$, where $\mathcal{S}$ and $\mathcal{A}$ denote the state space and the action space respectively, $\mathcal{T}(s'|s, a)$ describes the transition model, $r(s, a)$ gives the reward function, and $\gamma$ the discount factor. The goal of RL is to learn a policy $\pi(a|s)$ which maximizes the discounted expected gain defined as $\eta_M(\pi) = \underset{\pi, \mathcal{T}}{\mathbb{E}}[\sum_{t=0}^{\infty} \gamma^t r(s_t, a_t)]$.

Following the traditional transportation definitions listed in appendix B.1 , we now define the MDP of TSC, where an RL agent controls the traffic signal of an isolated intersection:

- **State:** The pressure, queue length, and running part in each traffic movement.
- **Action:** the traffic signal phase to be taken in the next interval.
- **Reward:** the average pressure of each traffic movement.

Under the offline setting, the policy $\pi$ is not allowed to interact with the real environment which enjoys the transition model $\mathcal{T}$ and learns from only a dataset $\mathcal{D}_M(s, a, s')$. The dataset is sampled by a trivial policy $\pi_B$ in $\mathcal{T}$. We can then define the problem of offline TSC as follows:

**Definition 1** (Offline TSC). *Given an offline dataset $\mathcal{D}_M(s, a, s')$ sampled from MDP $M(\mathcal{S}, \mathcal{A}, \mathcal{T}, r, \gamma)$ by a trivial policy $\pi_B$, the goal is to train a policy $\pi$ that maximize the following discounted expected gain:*

$$\eta_M(\pi) = \underset{\pi, \mathcal{T}}{\mathbb{E}}[\sum_{t=0}^{\infty} \gamma^t r(s_t, a_t)] \tag{1}$$

## 3 METHODOLOGY

In this section, we discuss the methodology of model-based offline TSC (MOTSC). Based on MOPO Yu et al. (2020), MOTSC is mainly distinguished by the well-crafted design of the transition model $\mathcal{T}'$. In the following subsections, we first illustrate the workflow of offline TSC. Then, we demonstrate the new MDP $\hat{M}(\mathcal{S}, \mathcal{A}, \mathcal{T}', \hat{r}, \gamma)$ constructed from the offline dataset $\mathcal{D}_M(s, a, s')$. After that, we further deep into the specific transition model we used in the new MDP $\hat{M}(\mathcal{S}, \mathcal{A}, \mathcal{T}', \hat{r}, \gamma)$ to learn the dynamics of TSC. Finally, we summarize the RL-based TSC method we exploit to parameterize the policy $\pi$.

### 3.1 WORKFLOW

Under the offline setting, TSC aims at training an offline control policy with an offline dataset whose data is sampled by trivial control policies (e.g. Fixed-time). As shown in Figure 1, we divide the workflow of MOTSC into three stages. The first stage samples offline data from the traffic environment. We use the Fixed-time Miller (1963) as the sampling policy when sampling data for the offline dataset, since it is the most common TSC policy used in real transportation management. In the second stage, we use the collected offline data to construct a new MDP $\hat{M}(\mathcal{S}, \mathcal{A}, \mathcal{T}', \hat{r}, \gamma)$. In the third stage, we exploit the new MDP $\hat{M}(\mathcal{S}, \mathcal{A}, \mathcal{T}', \hat{r}, \gamma)$ constructed in the second stage to train the offline policy. Since we decoupled the second and third stages, a wide range of online methods can easily fit into our workflow and change to offline methods.

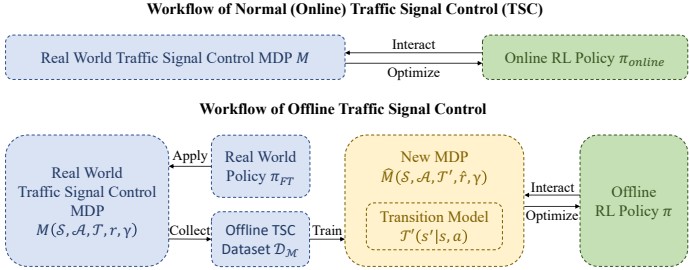

Figure 1: Comparison between workflows of normal traffic signal control and offline traffic signal control.

### 3.2 TACKLING THE OUT-OF-DISTRIBUTION OVER-OPTIMISM VIA MODEL-BASED TSC

In this section, we illustrate how the mitigation of the over-optimism Fujimoto et al. (2019) of out-of-distribution data can be naturally embedded into the construction of the MDP $\hat{M}(\mathcal{S}, \mathcal{A}, \mathcal{T}', \hat{r}, \gamma)$, which aligns well with the idea of model-based offline RL Yu et al. (2020); Kidambi et al. (2020); Yu et al. (2021). $\mathcal{T}'$ learns the transition dynamics from the offline dataset $\mathcal{D}_M$. We follow existing methods Yu et al. (2020; 2021) to parameterize $\mathcal{T}'$ with a Bayesian Neural Network (BNN) Neal (2012) $F$ with $N$ networks where we can leverage the deviation of the output of the N networks as the input of the uncertainty term $u(s, a)$ naturally. With this transition model, we can generate new data sample $(s, a, r, s')$ given a state $s$, and an action $a \sim \pi$, where we have the reward $r \sim r(s, a)$, the next state $s' \sim \mathcal{T}'(s'|s, a)$. These samples can be used as additional data to fill the blank out of the distribution of the offline dataset $\mathcal{D}_M$ in the training of policy $\pi$.

Towards conservative estimation of out-of-distribution samples, we follow MOPO Yu et al. (2020) to introduce a penalized term into the estimation of reward by transition model $\mathcal{T}'$. Specifically, the reward function $r(s, a)$ of the generated data is penalized by an uncertainty term $u(s, a)$:

$$\hat{r}(s, a) = r(s, a) - \lambda u(s, a) \tag{2}$$

where $\lambda$ is a hyperparameter controlling the scale of the penalty. We then use $\hat{r}$ instead of $r$ when generating new data samples. The final formulation of generated sample is $(s, a, \hat{r}, s')$ where we have the reward $\hat{r} = r - \lambda u$ with $r \sim r(s, a)$, and the next state $s' \sim \mathcal{T}'(s'|s, a)$.

We then give out the formulation of $u(s, a)$, the uncertainty term. Parameterized by BNN $F$, the transition model $\mathcal{T}'$ provides a straightforward estimation for the uncertainty $u(s, a)$ of the reward $r$ by the standard deviation of its $N$ outputs Lakshminarayanan et al. (2017).

$$u(s, a) = \sqrt{\frac{\sum_{i=1}^{N}(r_i(s, a) - E(r(s, a)))^2}{N - 1}} \tag{3}$$

With the transition model $\mathcal{T}'$, the penalized reward function, we can then define a new MDP $\hat{M}(\mathcal{S}, \mathcal{A}, \mathcal{T}', \hat{r}, \gamma)$. This new MDP is constructed from the offline dataset $\mathcal{D}_M$ and is available for policy $\pi$ to interact. We conduct $k$-rollout for policy $\pi$ that generates data for training the policy. The generated data $(s, a, \hat{r}, s')$ in the rollout consists of a new replay buffer $\mathcal{D}_{model}$. The policy $\pi(a|s)$ is finally trained on the MDP $\hat{M}$.

It's great to see that in TSC problems, inferring the reward from the next state directly is a feasible approach, as the definition of **State** and **Reward** in section 2 goes. So the reward function can be further expanded into $r(s, a) \sim s' \sim \mathcal{T}'(s'|s, a)$, where $f$ is a predefined and deterministic function to calculate the reward from the next state. So the main problem of constructing the new MDP $\hat{M}(\mathcal{S}, \mathcal{A}, \mathcal{T}', \hat{r}, \gamma)$ is the transition model $\mathcal{T}'$.

### 3.3 MOVEMENT INDEPENDENT TRANSITION: A SAMPLE-ENRICHED TRANSITION MODEL

As discussed before, the performance of model-based TSC depends highly on the performance of the transition model i.e. how well the transition model estimates the real traffic transition dynamics. However, existing transition models used in these methods (e.g. BNN Denker & LeCun (1990); MacKay (1992); Neal (2012)) cannot learn the traffic transition dynamics well, due to the complexity of it and the lack of model expressiveness. To tackle this challenge, we propose Movement Independent Transition, a novel transition model, to model the traffic transition dynamics. Movement Independent Transition enjoys explicit modeling of traffic transition dynamics. Furthermore, it promises a theoretical improvement in the accurate estimation of uncertainty, which is important in maintaining the exploration space for offline policies.

#### 3.3.1 MOVEMENT INDEPENDENT TRANSITION

Our intuition starts from the independent property of traffic movements in TSC. Let $k$ denote the number of traffic movements. The state $s \in \mathbb{R}^{\times k}$ and the action $a \in \mathbb{R}^{\times k}$ in TSC can be separated by the traffic movement denoted by subscripts, where $s_i$ indicates the number of vehicles in the traffic movements $i$ and $a_i \in \{0, 1\}$ indicates whether the traffic movement $i$ is allowed to pass the intersection.

**Theorem 1** (Independence of the Transition Model in TSC). *The Transition Model $T'(s'|s, a)$ in TSC can be expressed by the concatenate of the respective transitions of each traffic movement:*

$$\mathcal{T}'(s'|s, a) \leftarrow \underset{i=1,\dots,k}{Concat}[\mathcal{T}'(s_i'|s_i, a_i, i)] \tag{4}$$

The detailed proof can be found in the appendix. Based on the movement independence of the transition model, we separate the sample $(s, a, s') \in \mathcal{D}_M$ into $k$ samples $(s_i, a_i, s_i')$, by which we enrich $\mathcal{D}_M$ to a new dataset $\mathcal{D}'_{\hat{M}}$ with $k$ times of samples. The transition $\mathcal{T}'(s_i'|s_i, a_i, i)$ is then trained in $\mathcal{D}'_{\hat{M}}$.

To summarize, there are two intuitive advantages of using the movement independent transition as the transition model. First, it exploits an explainable function to infer the reward and the next state, which well fits the traffic transition dynamics. Second, it greatly enriches samples in the offline dataset by separating one sample into $k$ samples based on traffic movement, making it easy for the model to learn the transition dynamics.

#### 3.3.2 MOVEMENT INDEPENDENT TRANSITION ESTIMATES UNCERTAINTY ACCURATELY

Next, we study the theoretical guarantee that replacing the normal transition model $\mathcal{T}'(s'|s, a)$ with the movement independent transition $\underset{i=1,\dots,k}{Concat}[\mathcal{T}'(s_i'|s_i, a_i)]$ will gain more accurate estimation of

uncertainty, which helps solve the second challenge stated in the introduction. We note that $u(s, a)$ estimates the uncertainty of the sample $(s, a)$. Overestimation of the uncertainty happens when $u(s, a)$ does not fully exploit the dataset and prevents the algorithm from exploration. This decreases the lower bound of the optimal policy.

**Theorem 2** (Uncertainty overestimation decreases the lower bound). *With two uncertainty estimators $u(s, a)$ and $v(s, a)$, we denote policies trained with them by $\hat{\pi}_u$ and $\hat{\pi}_v$, respectively. If $u(s, a) \leq v(s, a)$, we have*

$$\inf_{\hat{\pi}_u}\{\eta_M(\hat{\pi}_u)\} \geq \inf_{\hat{\pi}_v}\{\eta_M(\hat{\pi}_v)\} \tag{5}$$

The detailed proof can be found in the appendix. Then, we consider the difference of $u(s, a)$ between $\mathcal{T}'(s'|s, a)$ and $Concat[\mathcal{T}'(s'_i|s_i, a_i, i)]$ in TSC. It has proven in theorem 1 that both forms represents the transition well with strong expressiveness of the used function. In practice, a Bayesian Neural Network $F$ is adopted. We then summarize the assumptions used in the following inference.

Given the dataset $\mathcal{D}_M$ and $\mathcal{D}'_M$, we can formulate the transition training as an interpolation problem, where we use the neural network $F$ to interpolate both $(s, a, s') \in \mathcal{D}_M$ and $(s_i, a_i, s'_i) \in \mathcal{D}'_M$, denoted by $F_1$ and $F_2$ respectively. Our analysis will lay on the assumption that both $F_1$ and $F_2$ resolves the interpolation perfectly with no uncertainty.

**Assumption 1.**

$$\mathcal{T}'_{F_1}(s'|s, a) = 1, (s, a, s') \in \mathcal{D}_M$$
$$\mathcal{T}'_{F_2}(s'_i|s_i, a_i) = 1, (s_i, a_i, s'_i) \in \mathcal{D}'_{\hat{M}} \tag{6}$$

This assumption is theoretically hold by theorem 1 and can be achieved by carefully crafting the model and conducting the training. We then discuss the Lipschitz continuity of $F$. The interpolation functions $F_1$ and $F_2$ adopted in $\mathcal{T}'(s'|s, a)$ and $\mathcal{T}'(s'_i|s_i, a_i)$ are assumed to be $K$-Lipschitz, which share the same Lipschitz constant.

**Assumption 2.**

$$F_1(x_1) - F_1(x_2) \leq K|x_1 - x_2|$$
$$F_2(x_1) - F_2(x_2) \leq K|x_1 - x_2| \tag{7}$$

**Theorem 3** (Upper bound of the uncertainty for the whole reward). *With assumption 1 and 2, we can write down an upper bound of the uncertainty $u(s, a)$ estimated by $F_1$ and $F_2$ and the upper bound of it. The detailed proof can be found in the appendix.*

$$u_{F_2}(s, a) = \frac{1}{k}\sum_{i=0}^{k-1} u_{F_2}(s_i, a_i)$$
$$\leq \frac{K}{n^{\frac{1}{2}}}\frac{1}{k}\sum_{i=0}^{k-1}||(s_i, a_i) - (s_i^i, a_i^i)||_2 \tag{8}$$

With upper bounds of the uncertainty under $F_1$ and $F_2$, we finally give the main theorem which provides theoretical guarantees for using the movement independent transition $F_2$:

**Theorem 4** (Movement Independent Transition decreases uncertainty overestimation). *Under assumption 1 and 2, the upper bound of the uncertainty with the movement independent transition $F_2$ is less than that of the normal transition $F_1$.*

$$\sup\{u_{F_2}(s, a)\} \leq \sup\{u_{F_1}(s, a)\} \tag{9}$$

The detailed proof can be found in the appendix. Theorem 4 points out that the movement independent transition $F_2$ holds a tighter upper bound for the uncertainty. While both transitions work well on the interpolation (assumption 1) of the offline dataset, $F_2$ reduces the overestimation of uncertainty and thus provides more space for the algorithm to explore.

Intuitively, using the movement independent transition guarantees that the estimated uncertainty is smaller than that by using BNN, the widely-used transition model. This relieves the common overestimation of uncertainty among offline RL and gives the offline policy more exploration space, which helps tackle the second challenge in the introduction.

### 3.4 SUMMARY OF MOTSC

By reparameterizing the transition model $\mathcal{T}'$ in the model-based TSC framework with Movement Independent Transition, we can finally summarize our policy optimization of MOTSC into the overview of the algorithm demonstrated in Figure 2.

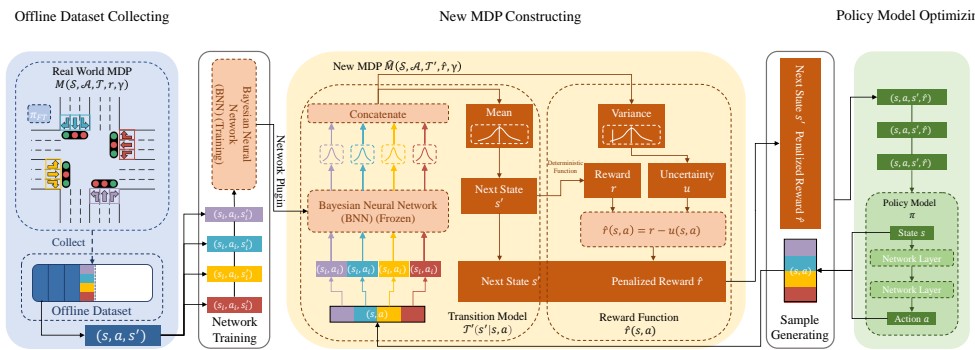

Figure 2: The training process of MOTSC. In the dataset sampling stage, we sample offline data of the intersection from the real world. In the transition model training stage, we use the offline dataset sampled before to train a transition model. In the policy optimizing stage, we let the agent interact with the transition model trained before to do exploration.

In the implementation, we use Advanced-MPLight Zhang et al. (2022) as the model of policy $\pi$. Based on MPLight Chen et al. (2020), Advanced-MPLight is the state-of-the-art RL-based TSC method. It separates the state into sub-states by traffic movements and learns a representation for the sub-state of each traffic movement. The signal control policy then makes decisions according to these representations. Note that Advanced-MPLight also tries to learn some movement-independent representations for TSC. This idea fits our transition model well so we use Advanced-MPLight as the policy $\pi$. For the RL algorithm, we follow the instruction in the paper of Advanced-MPLight and use Q-Learning Mnih (2015).

## 4 EXPERIMENTS

### 4.1 EXPERIMENTAL SETTINGS

In our experiments, we use traffic simulation by Cityflow Zhang et al. (2019) to simulate the real traffic flow, following existing studies Chen et al. (2020); Oroojlooy et al. (2020); Wu et al. (2021a;b); Zhang et al. (2022) in TSC. For traffic datasets, we use two datasets, Jinan, and Hangzhou, from the widely-used open-source traffic signal benchmark .etc (2023); Wei et al. (2019b;c). To better show the difference between different methods, we double the density of the traffic flow in these datasets.

In the offline data collecting stage, we select the minimum traffic signal cycle as 30 seconds, and the total time as 3600 seconds. We set a 5-second yellow signal. The number of epochs is set as 70. We do not use any online data in the training stage.

**Compared methods:** We compare our method with three groups of baseline methods. Transportation-based methods: methods based on transportation and not involving learning, including Fixed-time (FT) Miller (1963), MaxPressure (MP) Varaiya (2013), and Self-Organized Traffic Lights (SOTL) Cools et al. (2013). Offline RL-based methods: The only TSC problem natured method: ADAC-MDP Kunjir & Chawla (2022). Migrated methods: MOPO Yu et al. (2020), EDAC An et al. (2021), and IQL Kostrikov et al. (2021a). TSC methods: methods migrated from TSC methods and trained these methods on our offline dataset, including PressLight Wei et al. (2019a), AttendLight Oroojlooy et al. (2020), DQN Mnih (2015), CoLight Wei et al. (2019b), and Advanced-MPLight (A-MPLight) Zhang et al. (2022).

**Offline Datasets:** We use CityFlowZhang et al. (2019) to generate offline datasets, where the setting keeps the same as stated in Section 4.1. We use Fixed-time Miller (1963) policy as the sampling policy $\pi_B$ to control the traffic phase, for it is the most common TSC policy used in real urban traffic. Each sample includes a tuple $(s, a, s')$ of the state, the action, and the next state. The reward $r$ can be computed with $s'$. $s$ involves several features: traffic pressure Varaiya (2013), advanced-pressure Zhang et al. (2022), queue length, the number of running vehicles, and the topological relation of intersections. These features are picked according to the specific requirements of different methods.

**Evaluation Metrics:** We choose three representative metrics: pressure (Pres), queue length (QL), and waiting time (Wait), for evaluation. Pressure is the difference between waiting vehicles in the upstream and downstream lanes. Queue length is the number of vehicles waiting in front of a red traffic signal. Waiting time is the time for one vehicle from stopping to leave the intersection. These three metrics can comprehensively show the intersection of information and judge performance from space and time.

## 4.2 OVERALL PERFORMANCE

Table 1 shows the performance of all methods. We can see in both Jinan and Hangzhou, MOTSC outperforms all other offline RL methods and traditional transportation methods in all metrics.

Table 1: Comparison Result. Traditional; Offline; offline version of SOTA online models. The result consists of the mean value and $3\sigma$ (95% confidence probability)

| Dataset | Jinan (Small: 12 intersections) | | | Hangzhou (Medium: 16 intersections) | | | New York (Large: 196 intersections) | | |
|---|---|---|---|---|---|---|---|---|---|
| Methods | Pres↓ | QL↓ | Wait↓ | Pres↓ | QL↓ | Wait↓ | Pres↓ | QL↓ | Wait↓ |
| FT | 13.05 | 24.74 | 245.34 | 7.25 | 12.00 | 193.10 | 3.53 | 7.16 | 368.33 |
| MP | 6.16 | 12.88 | 134.47 | 5.66 | 7.33 | 123.72 | 3.50 | 6.39 | 240.25 |
| SOTL | 18.47 | 36.43 | 358.27 | 12.88 | 20.07 | 348.57 | 3.98 | 6.40 | 549.57 |
| ADACMDP | 15.95±0.00 | 22.61±0.00 | 897.98±0.00 | 8.53±0.00 | 12.20±0.00 | 597.79±0.00 | **2.88±0.00** | **3.34±0.00** | 266.87±0.00 |
| MOPO | 5.66±0.35 | 13.28±0.90 | 132.54±4.67 | 5.55±0.66 | **7.07±0.80** | 124.16±10.30 | 3.33±0.26 | 6.42±0.47 | 237.57±58.28 |
| EDAC | 32.67±0.00 | 55.43±0.00 | 711.49±0.00 | 10.64±0.00 | 24.27±0.00 | 403.37±0.00 | 4.27±0.00 | 5.72±0.00 | 870.59±0.00 |
| IQL | 30.81±0.11 | 49.31±0.35 | 1204.02±29.90 | 15.86±0.14 | 21.92±0.25 | 842.18±73.58 | 4.15±0.03 | 6.40±0.05 | 986.33±136.56 |
| PressLight-o | 26.33±3.74 | 46.97±3.59 | 725.68±133.38 | 7.96±0.81 | 12.08±1.97 | 195.15±26.13 | 4.08±0.37 | 5.89±0.17 | 514.12±86.39 |
| AttendLight-o | 18.94±4.86 | 33.48±7.27 | 388.43±138.12 | 6.93±0.72 | 8.42±0.53 | 139.95±5.72 | 3.69±0.27 | 6.81±0.36 | 308.42±26.60 |
| DQN-o | 20.65±5.94 | 37.33±6.71 | 351.49±114.09 | 13.37±0.82 | 17.09±0.49 | 483.68±40.36 | 4.33±0.18 | 6.65±0.25 | 859.12±117.08 |
| CoLight-o | 30.47±2.86 | 49.24±3.33 | 911.74±222.37 | 13.64±3.30 | 20.89±4.76 | 451.97±212.00 | 4.26±0.32 | 6.51±0.30 | 406.36±54.69 |
| A-MPLight-o | 18.07±7.21 | 29.71±11.32 | 328.87±143.04 | 6.49±0.69 | 8.48±0.78 | 138.29±8.84 | 3.66±0.46 | 6.62±0.99 | 240.56±76.56 |
| **MOTSC** | **5.39±0.36** | **12.81±1.58** | **124.61±5.50** | **5.26±0.52** | **7.07±0.50** | **119.49±5.44** | 3.31±0.30 | 6.32±0.43 | **230.73±77.31** |

As we can see, there is a huge performance discount if we simply change these SOTA online RL based TSC methods into offline versions. Without some specific algorithm to solve out-of-distribution problem, these methods cannot perform well in complex traffic conditions.

MOTSC performs well in the offline setting. As they perform better than SOTA traditional transportation method max pressure and the most commonly used method fixed time, it is an out-of-the-box benefit to implement these offline methods in the real world.

ADACMDP performs well under the New York dataset, but not as well under the Jinan and Hangzhou datasets. At the same time, we find all methods' performances are less varied in the New York dataset due to lower traffic flow density compared to the other datasets, but reproducing the same traffic flow density in such a large road network would result in unacceptable time costs in traffic simulator.

From figure 3 we can see those offline RL methods' performance under the different sizes of datasets. MOTSC outperformed all other offline methods in pressure, queue length and waiting time metrics. As the dataset size grows, unlike other methods, MOTSC's performance becomes better, and its standard deviation becomes smaller.

Moreover, MOTSC has larger advantages when the dataset size is small. With movement independent transition algorithms, MOTSC has a 12 times bigger dataset than other methods (Each intersection has 12 lanes). In the real world, effective data is precious, this feature makes MOTSC easy to implement in the real world.

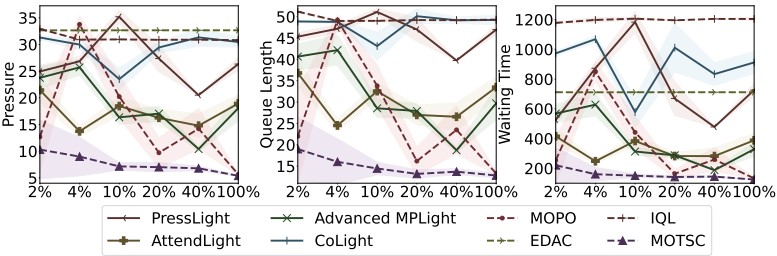

Figure 3: Test different offline RL with different dataset sizes. All these metrics are the smaller, the better. As offline dataset size grows, the standard deviation and mean of all metrics of MOTSC is decreasing, which indicates the model becomes better and more stable.

## 4.3 ABLATION STUDY

In the base MOTSC method, the policy model is adapted from Advanced MPLight. Actually, as a transition model, it can help migrate the online RL method to offline RL and improve their performance compared to other offline RL methods. So we tried using MOTSC as the fake environment instead the real environment most online RL based TSC used.

Figure 4 reports the three metrics achieved by all online RL based TSC methods with the different offline migrating methods. We can see all of these methods with MOTSC achieved better performance than simply migrating these methods to offline RL.

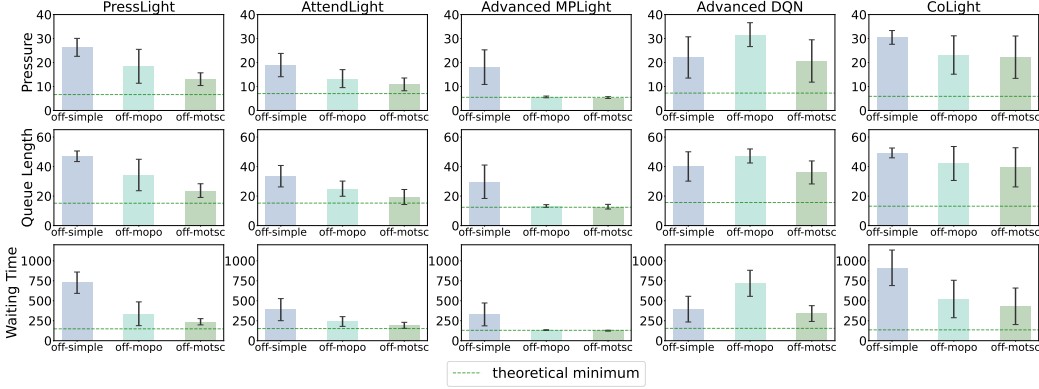

Figure 4: Offline method with the different online algorithm. All the metrics are smaller, the better. The green dash line is the theoretical minimum, which is the performance of the models trained in the real world. Off-simple is putting offline dataset into a replay buffer directly, and off-mopo or off-motsc is using MOPO or MOTSC to build a fake environment and letting different agents interact with it. The result consists of the mean value and $3\sigma$ (95% confidence probability)

## 4.4 CASE STUDY

**Validation of Transition Models** Comparing the accuracy of MOPO and MOSTSC transition model according to the equation $s' \sim \mathcal{T}'(s'|s, a)$, which means giving the model current step and action, comparing the next state in the real world and the next state the model predicted. Drawing the scatter and the results are shown in figure 5.

The more precise the transition model is, the better the fake environment can simulate the real world.

**Network design helps save samples** Figure 6 shows the distribution of the inputs of different network designs. With the same dataset, the scatter of MOTSC is more intensive under all the sizes of the dataset and the manifold area covered by the sample in MOTSC is significantly larger.

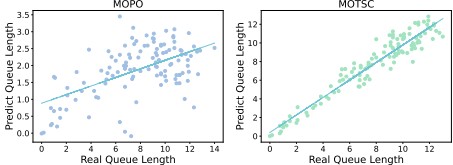 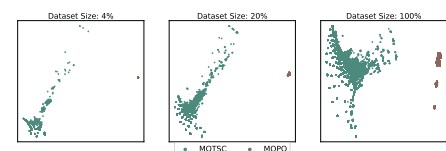

Figure 5: MOTSC transition model accuracy. The x-axis is the real queue length in the next state, and y-axis is the queue length in the next state predicted by the transition model. The R-square of MOPO is 0.32 and the R-square of MOTSC is 0.92.

Figure 6: Evaluation of offline dataset distribution with different network design. Both methods use the same offline dataset, MOTSC network design makes input data 12 times bigger than MOPO network's, and coverage becomes larger.

MOPO views the intersection as a whole, while MOTSC views all lanes in the intersection separately, using the state of one lane to train and predict. So with the same offline dataset size, MOTSC has bigger training samples than MOPO.

**Visualization of MOTSC and Max Pressure Policy** To have a better understand of why MOTSC performs better, we choose Max Pressure Varaiya (2013) to compare with, whose policy is easy to understand and has a good performance among other methods, as figure 7 shows.

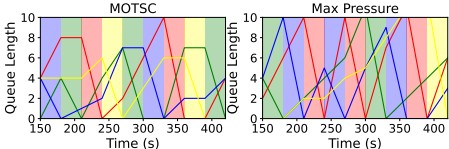 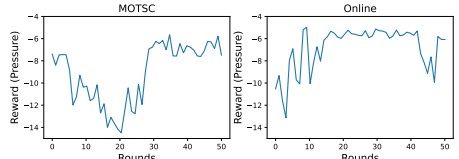

Figure 7: The visualization of MOTSC and Max Pressure. The background color means the traffic light phase, and the line means the number of vehicles allowed to pass under corresponding phase.

Figure 8: The comparison of training efficiency between a online method (Advanced MPLight) and its MOTSC offline version.

We observe that the max pressure method simply selects the lane with the highest pressure, without considering the running vehicles on it or estimating how the traffic flow will change over time. In contrast, MOTSC has learned to leverage these strategies. For example, at the 200th second, the "red lane" had the longest queue, but MOTSC did not it. Instead, it chose the "green phase" because it foresaw that the traffic flow on the "red lane" would not accumulate. Conversely, although the green lane had a shorter queue length, a large number of vehicles were approaching, as seen between the 210th and 270th seconds. Therefore, MOTSC achieved a smaller queue length on average compared to the max pressure method.

**Training Efficiency** To evaluate the training efficiency of our approach, we generated a training curve that displays the number of rounds required for MOTSC to converge. The results are shown in the figure 8. Although the learned transition model may not perfectly resemble the true environment, which contributes to longer convergence times, the difference between these is acceptable.

## 5 CONCLUSION

In this paper, we study the TSC problem under the offline setting, which is essential for the deployment of RL-based TSC policies. We proposed a MOTSC, a model-based RL algorithm to train offline TSC policies. Extensive experiments are conducted to show that MOTSC outperforms existing RL-based TSC methods and offline RL methods in the problem of offline TSC.

In future works, we will generalize MOTSC to more different traffic scenarios and some large-scale traffic datasets. We hope to push forward the application of RL-based TSC methods by overcoming the disadvantages of online training and improving data utilization.

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

# A  PROOF OF THEOREMS

## A.1  THE INDEPENDENCE OF TRANSITION MODEL IN TSC

**Theorem** (Independence of the Transition Model in TSC). *The Transition Model $T'(s'|s,a)$ in TSC can be expressed by the concatenate of the respective transitions of each traffic movement:*

$$\mathcal{T}'(s^{`}|s,a) \leftarrow \underset{i=1,\ldots,k}{Concat}[\mathcal{T}'(s'_i|s_i,a_i,i)] \tag{10}$$

*Proof.* The queue length in each traffic movement consists of three terms. The first is the number of vehicles that leave the movement. This is determined solely by the action $a_i$ of the traffic movement. We can write it down as $f_a(a_i)$. The second is the number of vehicles that join the movement. This can be induced by the position of traffic movement $i$ and written as $f_p(i)$. The third is the current queue length $s_i$. We denote this factor as $f_s(s_i)$. Therefore, the queue length of traffic movement $i$ can be formulated as:

$$f_{que} = f_p(i) + f_s(s_i) + f_a(a_i) \tag{11}$$

Similarly, we can prove that the running part and the pressure submit to the same independence property. The transition model is then expressed as the concatenation of $f$:

$$\mathcal{T}'(s'|s,a) \leftarrow \underset{i=1,\ldots,k}{Concat}[\mathcal{T}'(s'_i|s_i,a_i,i)] \leftarrow [f_{que}, f_{run}, f_{pres}] \tag{12}$$

Following the idea of $r(s,a) \sim s' \sim \mathcal{T}'(s'|s,a)$, the reward function can be expressed as:

$$r(s,a) = \frac{1}{k}\sum_{i=1}^{k} r_i(s_i,a_i,i) \tag{13}$$

$\square$

## A.2  UNCERTAINTY OVERESTIMATION DECREASES THE LOWER BOUND

**Theorem** (Uncertainty overestimation decreases the lower bound). *With two uncertainty estimators $u(s,a)$ and $v(s,a)$, we denote policies trained with them by $\hat{\pi}_u$ and $\hat{\pi}_v$, respectively. If $u(s,a) \leq v(s,a)$, we have*

$$\inf_{\hat{\pi}_u}\{\eta_M(\hat{\pi}_u)\} \geq \inf_{\hat{\pi}_v}\{\eta_M(\hat{\pi}_v)\} \tag{14}$$

*Proof.* Note that $u(s,a)$ describes the uncertainty between the real transition $\mathcal{T}$ and the learned transition $\mathcal{T}'$. We define $\epsilon_u(\pi)$ as the expected discount uncertainty with policy $\pi$:

$$\epsilon_u(\pi) = \underset{\pi,\mathcal{T}'}{\mathbb{E}}\big[\sum_{t=0}^{\infty}\gamma^t u(s_t,a_t)\big] \tag{15}$$

The lower bound of the trained policy $\pi$ is then given by Eq.16 (See the proof in MOPO Yu et al. (2020)).

$$\eta_M(\pi_u) \geq \eta_M(\pi) - 2\lambda\epsilon_u(\pi) \tag{16}$$

As stated in MOPO Yu et al. (2020), we also omit the dependence of $\epsilon_u(\pi)$ and $\epsilon_v(\pi)$ on $\mathcal{T}'$. With $u(s,a) \leq v(s,a)$, we have

$$\begin{aligned}
&\inf_{\hat{\pi}_u}\{\eta_M(\pi_u)\} - \inf_{\hat{\pi}_v}\{\eta_M(\hat{\pi}_v)\} \\
&= \sup_{\pi}\{\eta_M(\pi) - 2\lambda\epsilon_u(\pi)\} - \sup_{\pi}\{\eta_M(\pi) - 2\lambda\epsilon_v(\pi)\} \\
&= \sup_{\pi}\{2\lambda(\epsilon_v(\pi) - \epsilon_u(\pi))\} \\
&= \sup_{\pi}\{2\lambda\underset{\pi,\mathcal{T}'}{\mathbb{E}}\sum_{t=0}^{\infty}\gamma^t[v(s_t,a_t) - u(s_t,a_t)]\} \\
&\geq 0
\end{aligned} \tag{17}$$

$\square$

### A.3 UPPER BOUND OF THE UNCERTAINTY FOR THE WHOLE REWARD

**Theorem** (Upper bound of the uncertainty for the whole reward).

$$u_{F_2}(s,a) = \frac{1}{k} \sum_{i=0}^{k-1} u_{F_2}(s_i, a_i)$$

$$\leq \frac{K}{n^{\frac{1}{2}}} \frac{1}{k} \sum_{i=0}^{k-1} ||(s_i, a_i) - (s_i^i, a_i^i)||_2 \tag{18}$$

*Proof.* With assumption 1 and 2, we can write down an upper bound of the uncertainty $u(s,a)$ estimated by $F_1$ and $F_2$. Here, we only consider the reward function in the transition for simplification. For $r = F_1(s,a) = f(s'), s' \sim \mathcal{T}'(s'|s,a) : \mathbb{R}^{\times 2k} \to \mathbb{R}$, the output is limited within the range

$$F_1(s,a) \in [F_1(s^*, a^*) - K||(s,a) - (s^*, a^*)||_{2k},$$
$$F_1(s^*, a^*) + K||(s,a) - (s^*, a^*)||_{2k}]$$
$$(s^*, a^*) := \arg\min_{(s',a') \in \mathcal{D}_M} ||(s,a) - (s^*, a^*)||_{2k} \tag{19}$$

where $(s^*, a^*)$ is the nearest neighbor to $(s,a)$ in $2k$-dimensional space in the dataset $\mathcal{D}_M$. Estimated by Eq 3, we can give the upper bound of the uncertainty under $F_1$:

$$u_{F_1}(s,a) = \max_{i=1,\dots,N} ||\sum_{i=1}^{N} (s,a)||_F$$

$$\leq \left( \frac{K^2}{n} ||(s,a) - (s^*, a^*)||_{2k}^2 \right)^{\frac{1}{2}} \tag{20}$$

$$= \frac{K}{n^{\frac{1}{2}}} ||(s,a) - (s^*, a^*)||_{2k}$$

Then, we study the bound of the uncertainty under $F_2$ given by Eq 12. $F_2$ conducts interpolation by traffic movement $i$ respectively. We denote the movement by subscripts. With each function $F_{2,i}$ to be $k$-Lipschitz, we can write down the range

$$F_{2,i} \in [F_2(s_i^*, a_i^*) - K||(s_i, a_i) - (s_i^*, a_i^*)||_2,$$
$$F_2(s_i^*, a_i^*) + K||(s_i, a_i) - (s_i^*, a_i^*)||_2]$$
$$(s_i^i, a_i^i) := \arg\min_{(s',a') \in \mathcal{D}_M} ||(s_i', a_i') - (s_i^i, a_i^i)||_{2k} \tag{21}$$

Similar to Eq 20, we obtain the upper bound of the uncertainty for each movement under $F_2$:

$$u_{F_2}(s_i, a_i) \leq \frac{K}{n^{\frac{1}{2}}} ||(s_i, a_i) - (s_i^i, a_i^i)||_2 \tag{22}$$

Combining Eq 13 and Eq 22, we express the upper bound of the uncertainty for the whole reward as

$$u_{F_2}(s,a) = \frac{1}{k} \sum_{i=0}^{k-1} u_{F_2}(s_i, a_i)$$

$$\leq \frac{K}{n^{\frac{1}{2}}} \frac{1}{k} \sum_{i=0}^{k-1} ||(s_i, a_i) - (s_i^i, a_i^i)||_2 \tag{23}$$

$\square$

### A.4 MOVEMENT INDEPENDENT TRANSITION DECREASES UNCERTAINTY OVERESTIMATION

**Theorem** (Movement Independent Transition decreases uncertainty overestimation). *Under assumption 1 and 2, the upper bound of the uncertainty with the movement independent transition $F_2$ is less than that of the normal transition $F_1$.*

$$\sup\{u_{F_2}(s,a)\} \leq \sup\{u_{F_1}(s,a)\} \tag{24}$$

*Proof.* Two upper bounds are given by Eq 20 and Eq 23. Making use of the Triangle Inequality, we have

$$
\begin{aligned}
\sup\{u_{F_2}(s,a)\} &= \frac{K}{n^{\frac{1}{2}}} \frac{1}{k} \sum_{i=0}^{k-1} ||(s_i, a_i) - (s_i^i, a_i^i)||_2 \\
&\leq \frac{K}{n^{\frac{1}{2}}} \frac{1}{k} \sum_{i=0}^{k-1} ||(s_i, a_i) - (s_i^*, a_i^*)||_2 \\
&\leq \frac{K}{n^{\frac{1}{2}}} ||(s,a) - (s^*, a^*)||_{2k} \\
&= \sup\{u_{F_1}(s,a)\}
\end{aligned}
\tag{25}
$$

$\square$

# B  EXTENDED EXPERIMENT SETTINGS

## B.1  DEFINITIONS OF TRANSPORTATION CONCEPTS

We introduce some definitions in transportation to help formulate the MDP of traffic signal control:

- **Pressure:** Pressure is defined as the difference in vehicle density between entering and leaving the lane. The vehicle density of the lane is defined as $x(l)/x_{max}(l)$, where $x(l)$ is the number of vehicles in the lane $l$, $x_{max}(l)$ is the max number of vehicles in the lane $l$.

- **Queue Length:** Queue length is the number of waiting vehicles in front of an intersection.

- **Running Part:** Running part is the number of vehicles running in the lane. The sum of queue length and running part is the total number of vehicles in one lane.

- **Waiting Time:** Waiting time is the difference between leaving time and arriving time at the intersection for the vehicle.

- **Traffic Movement** Zheng et al. (2019): Traffic Movement is the traffic moving towards a certain direction. For an intersection, one traffic movement is combined with one entering lane and the corresponding leaving lane.

- **Movement Signal:** A movement signal regulates whether the vehicles in the traffic movement can go or not.

- **Phase:** Following the definition in Zheng et al. (2019), phase is the combination of movement signals.

## B.2  DATA SOURCES

The real traffic flow data comes from a widely-used open-sourced project .etc (2023). The data is based on camera data in those cities and necessary simplification has been done.

## B.3  DATA PREPROCESSING

Because of the low density of traffic flow in the original traffic flow file, the experimental results of most methods are close, so the performance differences between different methods cannot be tested effectively. Therefore, we increase the density of traffic flow. The generation of traffic flow is controlled by three parameters: start time, end time, and interval time. At present, the interval start time and end time of all traffic flows are the same, and the interval time is 1 second, so only one vehicle can be generated from one traffic flow. We tried to double, triple, and quadruple the traffic density by setting the end time to 1 second, 2 seconds, and 3 seconds after the start time. We find that when the traffic density reached 3 times higher than the original, none of the current methods could cope with the condition well. So we finally decide to set the traffic density to double the original traffic flow density.

## B.4 Network Design

The first layer of the transition model is a splitter that splits the state of an intersection into the 12 lanes. The second layer is a phase selector that divides the 12 lanes into two categories, based on whether their traffic lights are red or green, noting that all right turns are considered green. The third layer consists of two BNN networks, each of which predicts the next state for the two classes of lanes distinguished by selectors. The output result is passed through the ReLU activator to help reduce the training loss because the features of the intersection are always positive. The mean output of BNN is used to calculate the next state, and then use the next state to calculate the reward. The penalty is calculated through the standard deviation of BNN output, and the overestimation problem can be reduced by correcting the reward through penalty.

## B.5 Compared Methods

Transportation-based methods:

- Fixedtime Miller (1963): Fixed time control changes phase under a pre-determined cycle every fixed time plan. This control method is easy to conduct and has been widely used in most real-world intersections recently.
- SOTL Cools et al. (2013): Self-Organizing traffic signal Control can adaptively regulate traffic signals, which can achieve good performance by changing a few parameters.
- Maxpressure Varaiya (2013): Max pressure is an efficient traffic signal control method, which collects waiting vehicles in the upstream and downstream lanes and makes decisions.

Offline RL-based methods:

- MOPO Yu et al. (2020): This method uses an offline dataset to train a transition model to simulate a real environment, and calculate penalty according to multi-networks predict results to alleviate overestimation problem.
- IQL Kostrikov et al. (2021b): This method use known state-action pair to train so that it can avoid querying the values of unseen actions, ease the performance loss caused by out of distribution states.
- EDAC An et al. (2021): This method solves out of distribution problem by giving this sample high uncertainty instead of limiting agents exploring in distribution. Based on SAC-NAn et al. (2021), it decreases the number of Q-networks by increasing the gradient variety.

RL-based TSC methods:

- PressLightWei et al. (2019a): The traditional transportation based method max pressureVaraiya (2013) has achieved good performance. This method associates max pressure with reinforcement learning.
- AttendLightOroojlooy et al. (2020): This method is a policy-based model, which can be applied to most of traffic environments, including 3-way, 4-way intersections, and different kinds of roads.
- Advanced DQN: This method is based on DQN Mnih (2015), and adapted for traffic signal control problems.
- CoLightWei et al. (2019b): This method uses graph attentional networks to enable cooperation between different intersections, which performed well in large-scale road networks.
- Advanced MPLight Chen et al. (2020): Based on FRAP Zheng et al. (2019), this method takes both running part and queueing vehicles into account, to decide whether change the current phase, which is more efficient and reliable for deployment.

## B.6 Experimental Setups

In the offline dataset sampling stage. We use the traffic simulator CityFlow Zhang et al. (2019) to generate offline dataset. The input of CityFlow is the roadnet and the traffic flow accordingly. The roadnet files are $3 \times 4$ roadnet in Jinan and $4 \times 4$ roadnet in Hangzhou. The traffic flows are sampled from the real world called $anon\_3\_4\_jinan\_real\_2500$ and $anon\_4\_4\_hangzhou\_real\_5816$. The

density of these traffic flows have been doubled. There are some parameters to control the environment. Run counts, which means the total simulating time, is set to 3600 seconds. Interval, which is how much time for one step in the simulator means, is set to 1s. Min action time, which is the minimum time between two different phases. Yellow time is the time between the green light to the red light.

Our transition model has five FC layers. which has $I \times 16 \times 16 \times 16 \times O$ units, $I$ is the dimension of input tensor, and $O$ is the dimension of output output tensor. These layers have activation function swish, sigmoid, swish, swish, and ReLU accordingly. The last layer uses ReLU to model that all the traffic features are positive. The optimizer is Adam and the learning rate is $0.001$. The consistencies of state are pressure, queue length, and running part. We use a 4-phases action.

Our policy model has the same state and action settings as the transition model. There are some reinforcement learning related parameters, epochs are $80$, batch size is $20$. There is a replay buffer which the max size is $12000$, max sample size is $3000$. If the current number of samples is smaller than max sample size, samples all of them. The epsilon is set to $0.8$ with epsilon decay $0.95$, and the minimum epsilon is $0.2$. The loss function is mean square error.

