# OpenReview forum: "MOTSC: Model-based Offline Traffic Signal Control"
_ICLR.cc/2024/Conference — Submitted to ICLR 2024_

### Official Review · Reviewer_5yTR · 2023-10-29

**Soundness:** 3 good
**Presentation:** 3 good
**Contribution:** 3 good
**Rating:** 5
**Confidence:** 3

**Summary:**

The authors proposed an offline traffic signal control method based on model-based offline reinforcement learning. They formulated offline policy optimization under traffic signal control and designed the transition model. They also contained theoretical proof that the method can estimate the state of out-of-distribution samples more accurately. The paper also contains a presentation of experiments aiming to compare the introduced method with methods of traffic signal control and offline reinforcement learning under offline traffic signals.

**Strengths:**

- the method seems to be quite original in the domain of traffic signal control
- the experimental results are very good
- the paper is generally well-written (I found only some minor issues that should be fixed) and clear
- the quality of the paper is good, there is a good review of related works, a good explanation of the introduced methods, comprehensive and broad experiments carried out on 3 scenarios, an ablation study, proofs of theorems in the appendix

**Weaknesses:**

1. I did not find information about the availability of the code used in experiments, which limits the reproducibility of the results and their validation.
2. There are some minor writing issues that can be improved:
- p.2: "State: The pressure" -> "State: the pressure" (to be consistent with the next 2 points)
- p.3: would be good to give the full name of MOPO when it is first introduced (and the abbreviation in the brackets)
- p.5: "that both forms represents" -> "that both forms represent"
- p.5: "Given the dataset" -> "Given the datasets"
- p.5: "resolves the interpolation" -> "resolve the interpolation"
- p.5: "With assumption 1 and 2" -> "With assumptions 1 and 2"
- p.7: "Waiting time is the time for one vehicle from stopping to leave the intersection." - maybe "average time" is better?
- p.8: "online RL method to offline RL" -> "online RL methods to offline RL"
- p.8: "All the metrics are smaller, the better." -> "The smaller the metrics, the better."
- p.9: "To have a better understand" -> "To have a better understanding"
- p.9: "a online method" -> "an online method"
- p.9: "MOTSC did not it." - it seems that sth is missing here

**Questions:**

On p.5 there is a sentence "Overestimation of the uncertainty happens when u(s, a) does not fully exploit the dataset and prevents the algorithm from exploration.". Could you please explain it? I am not sure if it is clear enough, so might be good to add an explanation (or a reference).

---

> ### Author Response · Authors · 2023-11-23
>
> Thanks for your insightful feedback and for helping us find these writing issues, we will fix them as soon as possible.
>
> Q1: The uncertainty will be added as a penalty to the trained RL agent. When the RL agent wants to explore some unseen states, the trained model will penal this exploration since unlike the real world, the trained transition model does not really know how the unseen state will transit to another state, so we need to add the uncertainty as the penalty to prevent the agent from explore the unseen state unlimitedly. But if we overestimate the uncertainty, the penalty to the agent’s exploration will be too heavy and the agent will be too pessimistic[1].
>
> Hope these can solve your problems.
>
> [1] Why so pessimistic? Estimating uncertainties for offline rl through ensembles, and why their independence matters

---

### Official Review · Reviewer_o1af · 2023-10-31

**Soundness:** 2 fair
**Presentation:** 3 good
**Contribution:** 2 fair
**Rating:** 3
**Confidence:** 4

**Summary:**

The paper addresses the traffic signal control problem by introducing a model-based offline reinforcement learning approach. It presents the movement-independent transition, which can accurately estimate the uncertainty of the model from the offline dataset. Experiments validate that MOTSC outperforms existing RL-based traffic signal control methods, showing more stable and data-efficient traffic management.

**Strengths:**

The paper addresses the traffic signal control problem by introducing a model-based offline reinforcement learning approach. It presents the movement-independent transition, which can accurately estimate the uncertainty of the model from the offline dataset. Experiments validate that MOTSC outperforms existing RL-based traffic signal control methods, showing more stable and data-efficient traffic management.

**Weaknesses:**

1. The paper does not sufficiently explore the extensibility of its core concept, the "movement-independent transition," to alternative applications within offline RL. A more comprehensive examination of its adaptability to diverse offline RL algorithms would enhance the paper's practical utility.

2. The method uses information corresponding to 12 possible movements at the intersection, and it is not clear whether this is information that can be obtained in a real environment.

3. The paper heavily leans on the MOPO framework, both in terms of its structural framework and its theoretical underpinnings. A more comprehensive rationale for the selection of MOPO over other comparable alternatives is necessary.

4. The main method of the paper does not show a clear increase in performance compared to MOPO.  Also, the paper limits experimental evaluation to a single kind of offline dataset. This limited scope limits the generalizability of the results. Extending your assessment to include multiple datasets is critical to demonstrating real-world scalability.

**Questions:**

1. (From Weaknesses 1, 3) How about applying the movement-independent transition to another model-based offline RL algorithm?

2. (From Weaknesses 2) It is not obvious to use the information of 12 movements that are available in the intersection. How is this implemented? And, is it fair to compare with other methods that do not use this information?

3. (From Weaknesses 4) Are there any results of experiments on offline datasets obtained by other algorithms?

---

> ### Author Response · Authors · 2023-11-23
>
> Thanks for your insightful feedback.
>
> Q1: Since the “Movement Independent Transition” aims for the TSC problem according to its semantics and mathematical assumptions, it can be only applied to other model-based Offline TSC RL algorithms, and to the best of our knowledge, there is no other Offline RL algorithms aiming to TSC problem appeared in peer-reviewed conferences or journals. But if there are such a kind of algorithms in the future, we can apply them effectively since we first separate the intersection features into the granularity of lanes and do the training normally. Finally, we concatenate the output of the model from lane granularity to intersection granularity for RL agents making decisions.
>
> Q2: There are 12 lanes in one intersection (there are four directions, each direction has three lanes for left turn, straight, and right turn) so we can extract 12 movement information from one intersection. Since this is one of our main contributions, we didn’t apply it to other algorithms, but it can be conveniently added to other algorithms as Q1 described, making them perform better.
>
> Q3: The most common and simplest policy in the real world is fixed time policy. To make our method practical in the real world, we chose it as our sample policy collecting offline dataset. But since nowadays some traffic lights have deployed some better policies, e.g. max pressure policy, we can collect other offline datasets with these better policies in the future.
>
> Hope these can solve your problems.

---

### Official Review · Reviewer_JUsa · 2023-11-01

**Soundness:** 3 good
**Presentation:** 3 good
**Contribution:** 3 good
**Rating:** 6
**Confidence:** 3

**Summary:**

The paper proposed a model-based offline RL method for the traffic signal control problem. A transition model of the traffic flow is learned from offline dataset to replace the online interactive environment for the RL to learn traffic control policy. The authors proposed an appropriate definition for the MDP to restrict the complexity of the modeling. The uncertainty upper bound and out-of-distribution issues are also considered in the model learning using movement-independent transition. Empirically, the proposed algorithm outperforms multiple baselines in most of the simulation tasks.

**Strengths:**

1. The presentation is good and a lot of technical details are presented.
2. The quality is good, it is a good example of applying RL method to real-world large-scale decision-making problems. The experimental results are comprehensive and serve as a solid proof of the effectiveness of the proposed algorithm.
3. The f

**Weaknesses:**

1. More discussion on related works will be good for readers who are not very familiar with traffic signal control.
2. Some technical details are not explained clearly. For example, one major problem is the "Concat" in Theorem 1. What does the concat mean mathematically?
3. Assumptions might be too strict. For example, in assumption 2, the Bayesian neural networks are assumed to be Lipschitz. How to guarantee that in empirical implementations?

**Questions:**

1. See Weakness 2 & 3.
2. For the empirical evaluation metrics, you learned the offline RL policy on real-world dataset but evaluated the policy using a simulator, which does not look like a fair evaluation. How does the simulation compare to real-world traffic flows? Can you elaborate more on this?

---

> ### Author Response · Authors · 2023-11-23
>
> Thanks for your insightful feedback.
>
> Q1: “Concat” operation is an engineering problem. Since we proposed the concept of “Movement Independent Transition”, we separate the data in one intersection into 12 parts, each standing for a lane to decrease the error in uncertainty estimation while improving the offline data utilization. But for the RL agents, they need the whole intersection features vector as the input, thus we need to “Concat” the output of our model into the input of RL agents.
>
> Q2: We followed the assumption in the former work MOPO[1]. In our work, we use the transition model to mimic the real-world traffic flow, which helps the RL agents make better decisions. Since in the real world, the intersection features like queue length and pressure cannot change unlimitedly from time to time, that is why we assume the transition model, i.e. the Bayesian neural networks are K-Lipschitz.
>
> Q3: The real-world dataset are traffic flow dataset, which presents a vehicle’s trajectory from origin to destination, but how the vehicle interacts with other vehicles, with the environment, and with the traffic light is left to be unknown, we need both real-world dataset and simulator to fully simulate a vehicle’s behavior. Our policy is trained under the real-world traffic flow dataset combined with the simulator which deployed the simplest TSC policy: Fixed time policy. Using these data we have our offline RL agents trained. We evaluate also with real-world traffic flow dataset combined with the simulator but with our trained TSC policy. We can see the improvement compared to the sample policy: Fixed time policy.
>
> Hope these can solve your problems.
>
> [1] MOPO: Model-based Offline Policy Optimization, page 5

---

### Official Review · Reviewer_8gyt · 2023-11-01

**Soundness:** 1 poor
**Presentation:** 2 fair
**Contribution:** 1 poor
**Rating:** 3
**Confidence:** 4

**Summary:**

The authors design an offline reinforcement learning (RL) algorithm for a single traffic signal controller.

**Strengths:**

The authors present an offline reinforcement learning algorithm for TSC.

**Weaknesses:**

Please see the questions.

**Questions:**

The traffic signal control problem using deep reinforcement learning is quite an old problem now. Many solutions already exist in the literature that incorporate different multi-agent RL (MARL) techniques to solve this problem for a network of TSCs. The current paper only focuses on a single intersection and therefore, evades the rich interaction between different neighboring TSCs which lies at the heart of the problem.

1. The literature review is quite poor. The authors should better point out their novelty in the context of existing offline and online Deep RL-based TSC problems.
2. The techniques should be expanded to multiple TSCs.
3. Many commercial simulators are nowadays available that reliably mimic the traffic evolution of a network of TSCs, aiding the design of online algorithms.

---

> ### Author Response · Authors · 2023-11-23
>
> Thanks for your insightful feedback.
>
> Q1: This work has two main novelties. First, we study and formulate the problem of TSC under the offline RL setting. Although TSC is an old problem and has many novel methods, it still has a long distance to be implemented in the real world, and one of the main reasons is its unstable performance at the very early stage of online RL training, and what this work wants to do is to some extent make up for this problem. We adapt offline RL methods to TSC problem, using offline dataset train the agent, getting it prepared to be implemented in the real world. Second, we proposed the concept “Movement Independent Transition” and proved its effectiveness mathematically and experimentally in improving the model’s accuracy in estimating the uncertainty, which results in better prediction performance.
>
> Q2: Actually our method aims to multiple TSCs. All of the experiments shown in Table 1 are conducted under multiple TSCs settings. And the New York setting can even support intersections for up to 192.
>
> Q3: Making the simulators mimic the traffic flow more reliably and adapting offline methods  are two orthogonal approaches to the same problem, making the TSC agents more reliable when implementing it to the real world. However, both of these methods suffer from the problem of performance discount caused by distribution shift. This is because simulators cannot perfectly replicate real-world conditions, and offline datasets may not capture all the situations that can arise in the real world.
> To overcome the limitations of simulators, they need to be fine-tuned based on real-world data to accurately model vehicle flow and driving behavior. Unfortunately, none of the traffic signal control simulators currently available support fine-tuning of vehicle flow or calibration of driving behavior through the calibration API.
> To address the issues with offline RL, we can apply the uncertainty to the training process to improve the model’s generalizability. Both of these approaches hold significant potential, not just in traffic signal control but also in various fields that leverage RL.
> And our proposed methods followed the works in offline RL and adapted transportation theory to it to make is perform better under traffic signal control problem.
>
> Hope these can solve your questions.

---

### Meta-Review · Area_Chair_Uqgw · 2023-11-22

**Metareview:**

The paper suggests an offline RL approach using a model for traffic signal control. It replaces the online environment with a learned traffic flow model from a dataset, defining a suitable Markov Decision Process (MDP) to manage complexity. The model addresses uncertainty and out-of-distribution problems using a movement-independent transition. In simulations, the proposed algorithm consistently outperforms various baselines.

Strengths: The paper tackles traffic signal control using a model-based offline reinforcement learning method. It introduces movement-independent transition to accurately estimate model uncertainty from offline data. Experiments confirm that MOTSC surpasses current RL-based traffic signal control methods, demonstrating improved stability and data efficiency in traffic management.

Weaknesses:
1. Limited Scope: The paper focuses solely on a single intersection, neglecting the essential interaction between neighboring traffic signal controllers (TSCs) crucial for real-world traffic management. It fails to extend its techniques to address the broader challenge of coordinating multiple TSCs within a network.

2. Limited Literature Review: The literature review seemed insufficient, lacking a clear identification of the paper's novelty in the context of existing offline and online Deep RL-based traffic signal control problems.

3. Methodological and Theoretical Gaps: The assumptions need better justification. The method's core concept, the "movement-independent transition," is not thoroughly explored for adaptability to alternative offline RL applications, limiting practical utility. The reliance on the MOPO framework lacks a comprehensive rationale, and the absence of code availability hinders result reproducibility and validation. The paper's experimental evaluation is restricted to a single type of offline dataset, diminishing the generalizability of its findings.

**Justification For Why Not Higher Score:**

The reviewers are not as enthusiastic on the paper.

**Justification For Why Not Lower Score:**

N/A

---

### Decision · Program_Chairs · 2024-01-16

Reject